

# Importance of experimental information (metadata) for archived sequence data: case of specific gene bias due to lag time between sample harvest and RNA protection in RNA sequencing

Tomoko Matsuda

Nihon BioData Corporation, Kawasaki, Kanagawa, Japan

Corresponding author
Tomoko Matsuda,
tomokom@nbiodata.com

## ABSTRACT

Large volumes of high-throughput sequencing data have been submitted to the Sequencing Read Archive (SRA). The lack of experimental metadata associated with the data makes reuse and understanding data quality very difficult. In the case of RNA sequencing (RNA-Seq), which reveals the presence and quantity of RNA in a biological sample at any moment, it is necessary to consider that gene expression responds over a short time interval (several seconds to a few minutes) in many organisms. Therefore, to isolate RNA that accurately reflects the transcriptome at the point of harvest, raw biological samples should be processed by freezing in liquid nitrogen, immersing in RNA stabilization reagent or lysing and homogenizing in RNA lysis buffer containing guanidine thiocyanate as soon as possible. As the number of samples handled simultaneously increases, the time until the RNA is protected can increase. Here, to evaluate the effect of different lag times in RNA protection on RNA-Seq data, we harvested CHO-S cells after 3, 5, 6, and 7 days of cultivation, added RNA lysis buffer in a time course of 15, 30, 45, and 60 min after harvest, and conducted RNA-Seq. These RNA samples showed high RNA integrity number (RIN) values indicating non-degraded RNA, and sequence data from libraries prepared with these RNA samples was of high quality according to FastQC. We observed that, at the same cultivation day, global trends of gene expression were similar across the time course of addition of RNA lysis buffer; however, the expression of some genes was significantly different between the time-course samples of the same cultivation day; most of these differentially expressed genes were related to apoptosis. We conclude that the time lag between sample harvest and RNA protection influences gene expression of specific genes. It is, therefore, necessary to know not only RIN values of RNA and the quality of the sequence data but also how the experiment was performed when acquiring RNA-Seq data from the database.

## INTRODUCTION

High-throughput sequencing (HTS) has been applied in several fields of biological research since its appearance about a decade ago and has made it possible to collect data with substantially higher-throughput and lower-cost (*Kircher & Kelso, 2010*; *Kodama, Shumway & Leinonen, 2012*). Large volumes of HTS data have been submitted to the Sequence Read Archive (SRA, *Leinonen, Sugawara & Shumway, 2010*), and repository users can access the archived data for reuse. *Nakazato, Ohta & Bono (2013)* constructed a publication list including PubMed IDs (PMIDs) referring to SRA entries. This list is useful for retrieving archived data based on the experimental details (metadata); however, metadata described by the data depositor lack some important information about protocol steps (*Alnasir & Shanahan, 2015*). *Ohta, Nakazato & Bono (2017)* calculated the quality of all archived sequence data in the SRA to allow users to control not only metadata but also the data quality in their searches.

It may not be possible to clarify the validity of the sequence data by examining the quality control data generated by a quality control tool (*e.g.*, FastQC) without metadata containing experiment-specific information. For example, this author has obtained two RNA samples: one isolated from cultured cells in 30 min after sampling, and the other isolated from the same cells in 60 min after sampling; both of these RNA samples prepared in the same laboratory showed high RNA integrity number (RIN) values indicating non-degraded RNA. However, it remains questionable whether data obtained from RNA samples with different time lags from sample harvest to RNA protection can be handled equally. In the case of RNA sequencing (RNA-Seq), it is necessary to consider that changes in gene expression can occur over a short time period (several seconds to a few minutes) in many organisms. For instance, in the filamentous fungus *Aspergillus fumigatus*, 23% and 35% of genes were differentially expressed after 30 and 120 min of hypoxia exposure, respectively (*Losada et al., 2014*) compared with before exposure. In the case of cultured cells, dynamic changes of gene expression profiles occurred during the first 2 h (30, 60, 90, and 120 min) of serum stimulation in hTERT immortalized foreskin-derived human fibroblasts (*Kirkconnell et al., 2016*). Therefore, to isolate RNA that accurately reflects the transcriptome at the point of harvest, raw biological samples should be frozen in liquid nitrogen or treated with RNA stabilization reagent or RNA lysis buffer as soon as possible.

Some cultured mammalian cells need to be suspended in 1 × PBS to remove fetal bovine serum and/or media before treatment with RNA lysis buffer. For example, the strategy for RNA sample processing from suspended Chinese hamster ovary (CHO) cells at the author's laboratory at Nihon BioData Corporation is as follows: (1) The medium containing the suspended CHO cells is transferred from the cell culture flask to a 1.5 mL centrifuge tube. (2) The centrifuge tube is centrifuged for 5 min at 300×$g$ and all supernatant is removed. (3) 200 μL of 1 × PBS is added to the cell pellet and pipetted up and down five times to thoroughly suspend the pellet. (4) The centrifuge tube is centrifuged for 5 min at 300×$g$ and all supernatant is removed. (5) The centrifuge tube is

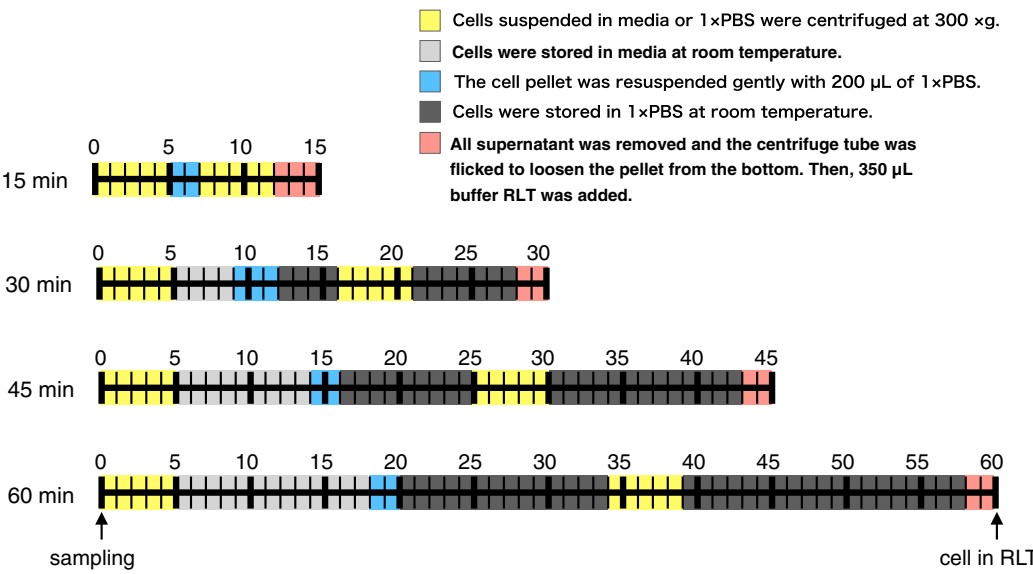

**Figure 1 Time course illustration of the experiment.** RNA in CHO cells was protected at different times (15, 30, 45, and 60 min) after cell harvest by the addition of RNA lysis buffer (buffer RLT, QIAGEN). Except for the 15 min samples, cells were left in media or 1 × PBS at room temperature for 4–13 min or 11–33 min, respectively.               

flicked to loosen the cell pellet from the bottom. (6) 350 μL of buffer RLT (RNA lysis buffer containing guanidine thiocyanate; QIAGEN, Hilden, Germany) is added to the pellet and pipetted up and down five times to mix thoroughly. (7) The entire cell suspension is immediately loaded onto a QIAshredder spin column (QIAGEN) and homogenized by centrifugation for 2 min in a microcentrifuge at maximum speed. When cells are treated by using the above strategy, the time lag until the addition of RNA lysis buffer, which contains guanidine hydrochloride, is critical. Because RNA in cells suspended in RNA lysis buffer immediately inactivates RNases to ensure isolation of intact RNA, the time lag until the addition of this reagent should be as short as possible. If the number of samples is less than 10, RNA lysis buffer can be added to the cell pellet within about 15 min. As the number of samples increases, however, the time it takes for centrifuging and pipette operation for samples becomes longer. As a result, the cells remain in media or 1 × PBS at the conditions (temperature and $CO_2$ concentration) outside the incubator for a long time.

Here, to evaluate whether RNA-Seq data obtained from RNA samples with a different lag time until RNA protection can be analyzed equally, we harvested CHO-S cells after 3, 5, 6, and 7 days of cultivation, added RNA lysis buffer at various lag times (Fig. 1; 15, 30, 45, and 60 min), purified the RNA, and then performed RNA-Seq. We found that the global trends of gene expression were similar, but the expression of some specific genes was significantly different between samples that were processed with different lag times until RNA protection on the same cultivation day. Most of the genes whose profiles changed with lag time until RNA protection were related to apoptosis. Based on our results, we propose to describe detailed protocol procedures including working time for reusing sequence data accessed from the data repository.

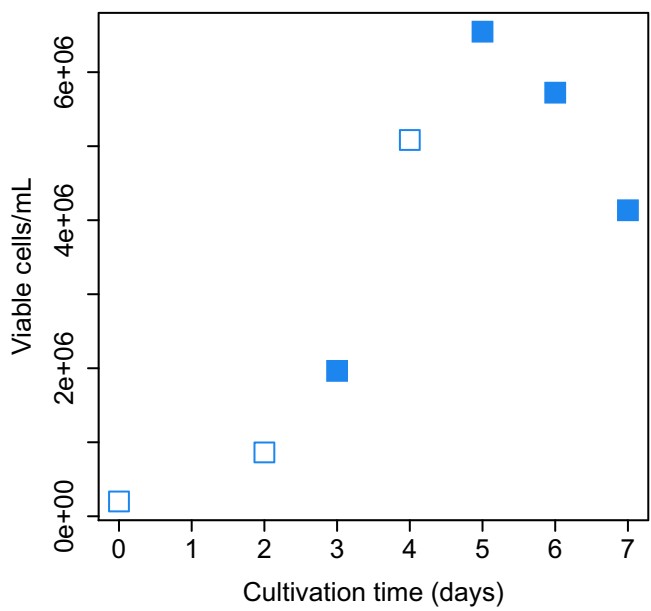

**Figure 2 Growth curve of CHO-S cells.** Cells used for the experiment are indicated by filled squares.

# MATERIALS & METHODS

## Cells

Bach CHO-S (Thermo Fisher Scientific, Waltham, MA, USA) cell culture was grown in 20 ml of Gibco CD-CHO medium (Thermo Fisher Scientific, Waltham, MA, USA) containing 400 μL of 200 mM L-Alanyl-L-glutamine (final concentration: 4 mM; Sigma–Aldrich, St. Louis, MI, USA) and 40 μL of Gibco Anti-Clumping Agent (final concentration, 0.2% (v/v); Thermo Fisher Scientific, Waltham, MA, USA) in a 125 mL flask. The flask was shaken (120 rpm) at 37 °C in a humidified 5% $CO_2$ atmosphere. Determination of total and viable cell numbers was performed with Vi-Cell XR Cell Viability Analyzer (Beckman Coulter, Fullerton, CA, USA) and CellProfiler software (*Carpenter et al., 2006*). The images obtained from Vi-Cell XR were analyzed and total and viable cell numbers were counted using the CellProfiler pipeline. The growth curve of CHO-S cells over the period of study is shown in Fig. 2.

## Harvesting samples, RNA isolation, cDNA library construction, and RNA-Seq

RNA in CHO cells was protected at different times (15, 30, 45, and 60 min) after harvest by the addition of RNA lysis buffer. Two persons performed this experiment: one person opened and closed the lids of the tubes, added reagent, and carried out the pipetting, and another put the tubes in and out of a microcentrifuge. The experiment was conducted as follows (Fig. 1). A total of three technical replicates for every sample were prepared. The medium containing the suspended CHO cells (400,000 cells) was transferred from the cell culture flask to 1.5 mL centrifuge tubes. Each centrifuge tube was centrifuged for 5 min at 300×*g* and all supernatant was removed. An aliquot of 200 μL of Gibco PBS

(1×, calcium and magnesium-free, pH 7.2; Thermo Fisher Scientific, Waltham, MA, USA) was added to the cell pellet and pipetted up and down five times to resuspend thoroughly. The centrifuge tube was centrifuged for 5 min at 300×$g$ and all supernatant was removed. The centrifuge tube was flicked to loosen the cell pellet from the bottom. An aliquot of 350 μL of the buffer RLT (RNA lysis buffer; QIAGEN, Hilden, Germany), which contains guanidine thiocyanate, was added to the pellet and pipetted up and down five times to mix thoroughly. The time it took to complete these steps was set to 15, 30, 45, or 60 min (Fig. 1). The entire cell suspension was immediately loaded onto a QIAshredder spin column (QIAGEN, Hilden, Germany) and homogenized by centrifugation for 2 min in a microcentrifuge at maximum speed. The resultant homogenized lysate was kept at −80 °C until RNA isolation. Total RNA isolation was performed using an RNeasy mini kit (QIAGEN, Hilden, Germany). The quality of total RNA was assessed by RIN values using the Agilent 2200 TapeStation (Agilent Technologies, Santa Clara, CA, USA). Libraries were prepared using a TruSeq Stranded mRNA Library Prep Kit (Illumina, San Diego, CA, USA). The finished cDNA libraries were quantified by using the Agilent 2200 TapeStation and sequenced on an Illumina NextSeq 500 platform with 75-bp single-end reads. The experiment procedures of this section were deposited to protocol.io with DOI: 10.17504/protocols.io.57ng9me.

## Data analysis

All the reads were deposited in the DDBJ Sequence Read Archive (accession number: DRA006016). The metadata associated with the data submission to DDBJ DRA006016 can be obtained from SRA Run Selector (https://www.ncbi.nlm.nih.gov/Traces/study/?acc=DRP004803&o=acc_s%3Aa). The quality of the raw reads was analyzed with FastQC (version 0.11.3; Andrews, 2010). All short reads were mapped to the CHO-K1 RefSeq assembly (22,516 sequences; RefSeq Assembly ID: GCF_000223135.1) and CHO-K1 mitochondrial DNA (1 sequence; RefSeq Assembly ID: GCF_000055695.1) using Bowtie2 (version 2.3.4.1; Langmead & Salzberg, 2012) and quantified using RSEM (version 1.2.31, Li & Dewey, 2011). Shannon's information entropy, which can be used to assess changes in transcriptome diversity (Ogata, Yokoyama & Iwabuchi, 2012; Ogata et al., 2015; Seekaki & Ogata, 2017; Kannan et al., 2020; Liu et al., 2020) was calculated based on TPM (transcripts per million). Principal component analysis (PCA) based on TPM was performed with the prcomp function with scale = TRUE in the R Stats package (version 3.3.3; R Development Core Team, 2016). A hierarchical cluster analysis (HCA) dendrogram based on TPM using hclust function in the R Stats package (version 3.3.3, R Development Core Team, 2016). The hclust clustering was performed using distance metric based on Spearman's rank correlation coefficient and clustering method "ward.D2". Differentially expressed gene (DEG) analyses were performed using the TCC package with edgeR on R console (Robinson & Oshlack, 2010; Sun et al., 2013). Genes with a false discovery rate (FDR) less than 0.05 were identified as differentially expressed. The identified DEGs were compared between 15 min and each other time period (30, 45, and 60 min) on each cultivation day and the numbers of common DEGs among the four different days were counted. Orthology of CHO-K1 RefSeq genes and genes associated

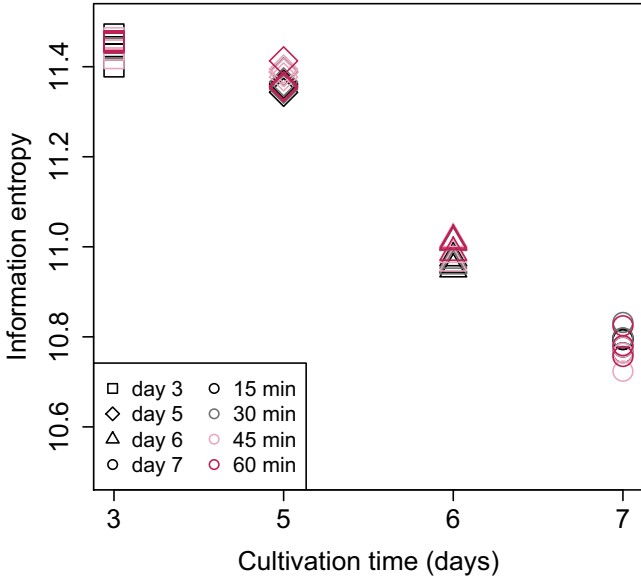

**Figure 3 Scatter plot of cultivation day vs. Shannon's information entropy.** There were no significant differences in the value of Shannon's information entropy between samples from the same cultivation day with different processing lag times.

with apoptosis (249 genes with GO terms included "apoptotic" or "apoptosis", Table S1) in the MGI (Mouse Genome Informatics, http://www.informatics.jax.org/) database was confirmed by using BLASTN searches (E-value of 1.0E-50 as a threshold). Then, the number of genes associated with apoptosis in the common DEGs was counted.

## RESULTS

To determine whether RNA-Seq data obtained from RNA samples that differ in the time lag from harvesting cells to protecting RNA (hereafter, "processing lag time") can be analyzed equally, we performed RNA-Seq of RNA isolated from CHO-S cells (at 3, 5, 6, and 7 days of cultivation) with RNA protected at various times (15, 30, 45, and 60 min) following cell harvest, as shown in Fig. 1. High-quality total RNA was obtained from all samples (RIN values > 9.1). We prepared cDNA libraries from total RNA and obtained 11 to 42 million sequence reads per sample. The mean base quality score in the Phred scale was 35.68 indicating that the base call accuracy was above 99.97%. Sequence mapping rates to reference sequences ranged from 78.19% to 84.29%.

Shannon's information entropy of RNA-Seq data of bach CHO-S cell culture was shown in Fig. 3. There were no significant differences in the value of Shannon's information entropy between samples from the same cultivation day with different processing lag times. On the other hand, the value of Shannon's information entropy decreased with cultivation. Decreasing Shannon's information entropy during the cultivation is also seen in other organisms used in industry (unpublished data).

PCA (Fig. 4) and HCA (Fig. 5) were conducted using RNA-Seq data mapped onto the CHO-K1 RefSeq assembly (GCF_000223135.1) and CHO-K1 mitochondrial DNA (GCF_000055695.1). PC1 (cumulative contribution ratio (%) = 0.37) appeared to be
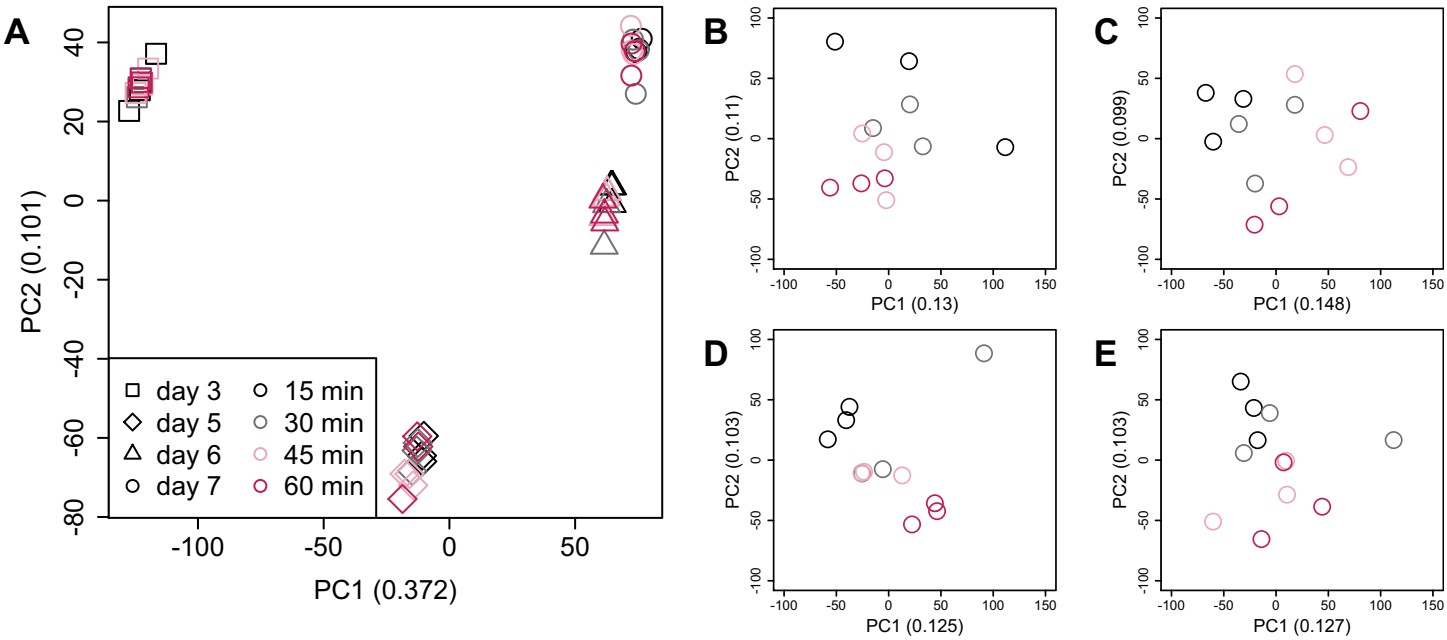

**Figure 4 Score plot of principal component analysis (PCA) using PC1 and PC2.** PCA plot of the 48 samples included in the study based on data from CHO-K1 RefSeq assembly (GCF_000223135.1) and CHO-K1 mitochondrial DNA (GCF_000055695.1) with at least one mapped read on at least one of each sample. PC1 appeared to be associated with cultivation day, but neither PC1 nor PC2 were associated with the processing lag time. (A) All 48 samples. Different colors identify different processing lag times, while different shapes identify different cultivation days. (B), (C), (D), and (E) Samples from day 3, 5, 6, and 7, respectively. Legend is the same as (A).

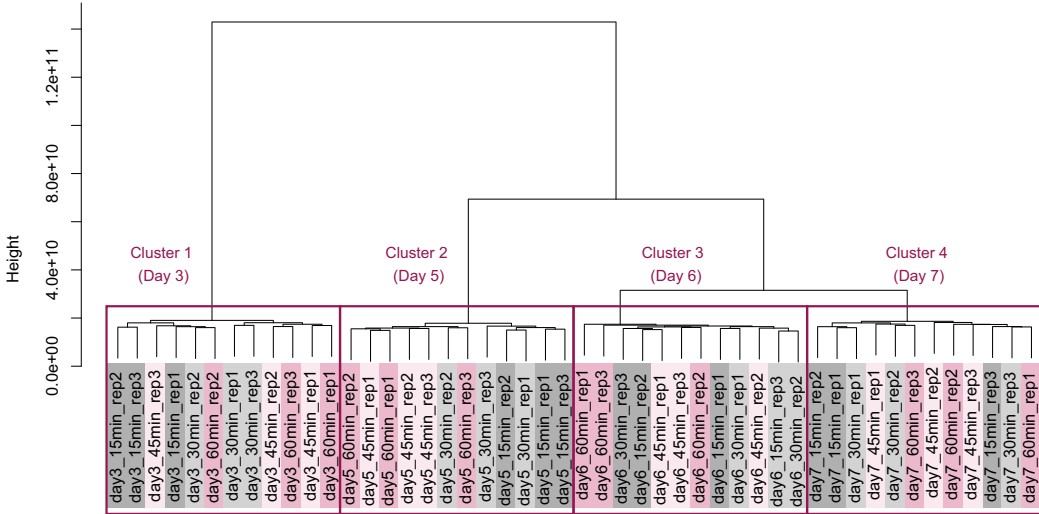

**Figure 5 Dendrogram resulting from hierarchical clustering analysis (HCA).** HCA was performed based on data from CHO-K1 RefSeq assembly (GCF_000223135.1) and CHO-K1 mitochondrial DNA (GCF_000055695.1) with at least one mapped read on at least one of each sample. Each sample is indicated by the number of days of cultivation, the number of minutes of the processing lag time, and the technical replicate number. The dendrogram showed that samples of the same cultivation day clustered together; there were four clusters associated with cultivation day.

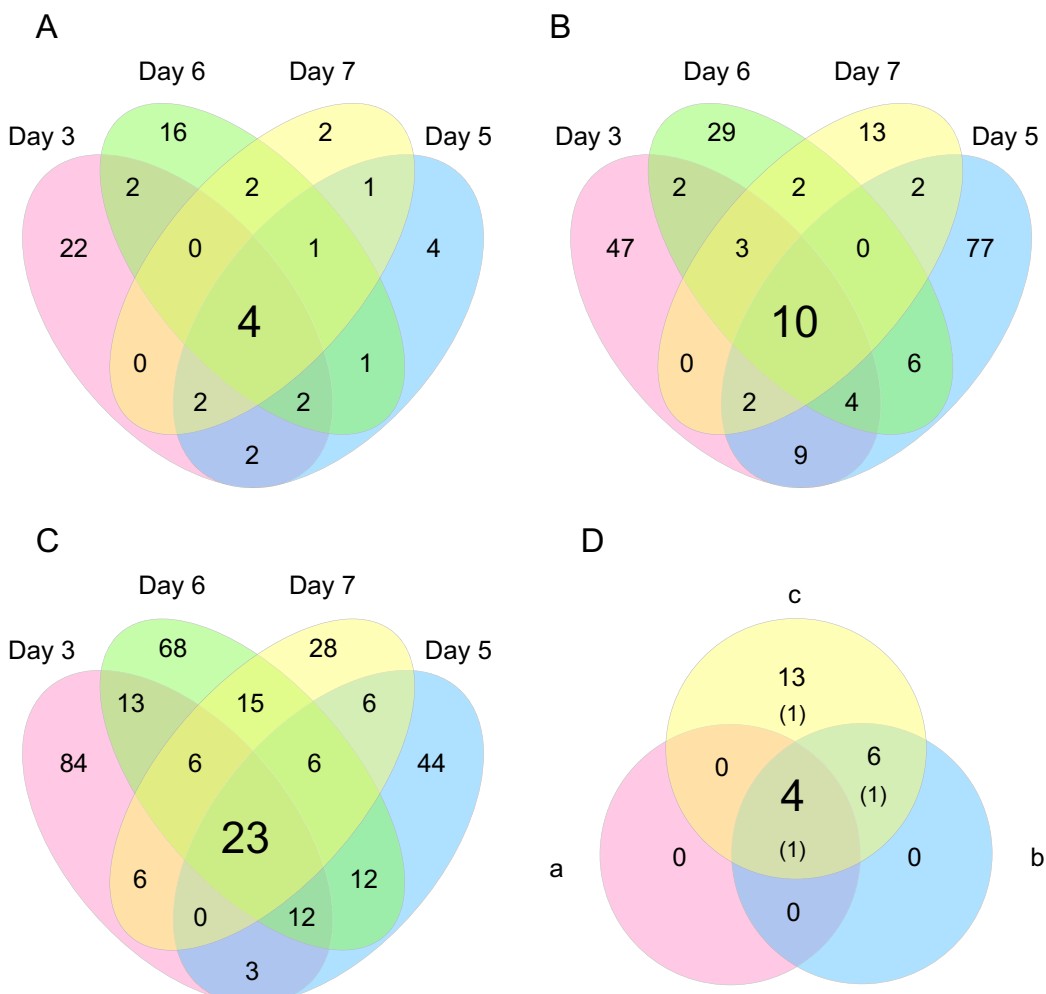

**Figure 6 Venn diagrams of differentially expressed genes (DEGs).** The numbers of genes that were DEGs between different processing lag times among all 4 different days were (A) 4 (15 min vs. 30 min), (B) 10 (15 min vs. 45 min), and (C) 23 (15 min vs. 60 min). (D) 4 common DEGs were detected among (a) 15 min vs. 30 min, (b) 15 min vs. 45 min, and (c) 15 min vs. 60 min.

associated with cultivation day (Fig. 4A), but neither PC1 nor PC2 (cumulative contribution ratio (%) = 0.10) were associated with the processing lag time (Figs. 4A–4E). The HCA dendrogram showed that samples of the same cultivation day clustered together; there were four clusters associated with cultivation day (Fig. 5). These results show that the gene expression was strongly associated with cultivation day, but the correlation between the gene expression and the processing lag time was not found. Even if processing lag times are different, global trends of gene expression were similar between samples from the same cultivation day.

Differentially expressed gene (DEG) analysis was conducted to explore the correlation between the processing lag time (15, 30, 45, and 60 min) and the gene expression. First, we compared the gene expression of RNA-Seq data between 15 min and each other time period (30, 45, and 60 min) on each cultivation day. The numbers of common DEGs

**Table 1  Common differentially expressed genes (DEGs) among four different days.**

| Gene name | CHO-Refseq | MGI ID | GO terms included "apoptosis" or "apoptotic" | Common DEGs among days | | |
|---|---|---|---|---|---|---|
| | | | | 15 min vs. 30 min | 15 min vs. 45 min | 15 min vs. 60 min |
| Atf4 | XM_007654285.1, NM_001246812.1 | MGI:88096 | GO:0043525, GO:0070059, GO:1905461 | – | – | + |
| Btg2 | XM_003505756.2 | MGI:108384 | GO:0043066, GO:0043524 | – | – | + |
| Chub2 | NM_001244378.1 | * | – | – | + | + |
| Cyr61 | XM_007653329.1, XM_003512942.2 | MGI:88613 | GO:0003278, GO:0043065, GO:0043066, GO:0043280 | – | – | + |
| Ddx5 | XM_003501860.2 | MGI:105037 | GO:0072332 | – | – | + |
| Dusp5 | XM_007653051.1 | MGI:2685183 | – | – | – | + |
| Egr1 | XM_003502541.2 | MGI:95295 | GO:0042981, GO:0043525 | + | + | + |
| Egr2 | XM_003515916.2 | MGI:95296 | – | – | + | + |
| Fos | NM_001246683.1 | MGI:95574 | – | + | + | + |
| Fosb | XM_007645972.1 | MGI:95575 | – | – | – | + |
| H3f3b | XM_003506831.2 | MGI:1101768 | – | – | – | + |
| Ier3 | XM_003505887.2 | MGI:104814 | GO:0008630, GO:0043066, GO:1901029 | – | + | + |
| Ier5 | XM_007640860.1 | MGI:1337072 | – | – | – | + |
| Jun | XM_007643818.1 | MGI:96646 | GO:0043065, GO:0043066, GO:0043524, GO:0043525 | – | + | + |
| Junb | XM_007642561.1 | MGI:96647 | – | + | + | + |
| Klf6 | XM_003494905.2, XM_003515975.2 | MGI:1346318 | – | – | – | + |
| LOC103159497 | XR_484059.1 | * | – | + | + | + |
| Plk2 | XM_003496201.2 | MGI:1099790 | GO:0043066, GO:0071866 | – | – | + |
| Ppp1r15a | XM_007649270.1 | MGI:1927072 | GO:0006915 | – | – | + |
| Sgk1 | XM_003511162.2 | MGI:1340062 | GO:0006915, GO:0043066 | – | – | + |
| Srsf5 | XM_003498255.2, XM_007625054.1, XR_480198.1 | MGI:98287 | – | – | + | + |
| Vegfa | XM_007642338.1 | MGI:103178 | GO:0043066, GO:0043154, GO:0043524, GO:2001237 | – | – | + |
| Zfp36 | XM_007644391.1 | MGI:99180 | GO:1902172 | – | + | + |

Notes:
An asterisk symbol (*) indicates that the gene was not included in the MGI (Mouse Genome Informatics) database.
A plus symbol (+) indicates that the gene was detected as a common DEG.
A minus symbol (–) indicates that the gene was not detected as a common DEG.

among the four different days were 4 (15 min vs. 30 min), 10 (15 min vs. 45 min), and 23 (15 min vs. 60 min) (Fig. 6). The number of common DEGs increased as the processing lag time increased (Fig. 6, Table 1). All 23 common DEGs, with the exception of LOC103159497, were upregulated in 30, 45, and 60 min samples compared with 15 min samples (Table 1, Fig. S1). A total of 12 of the 23 common DEGs were orthologous to genes associated with apoptosis in the MGI database (Table 1).

PCA and HCA was then performed for genes identified as differentially expressed at least once in DEG analyses. In the PCA plot, PC1 and PC2 appeared to be associated with

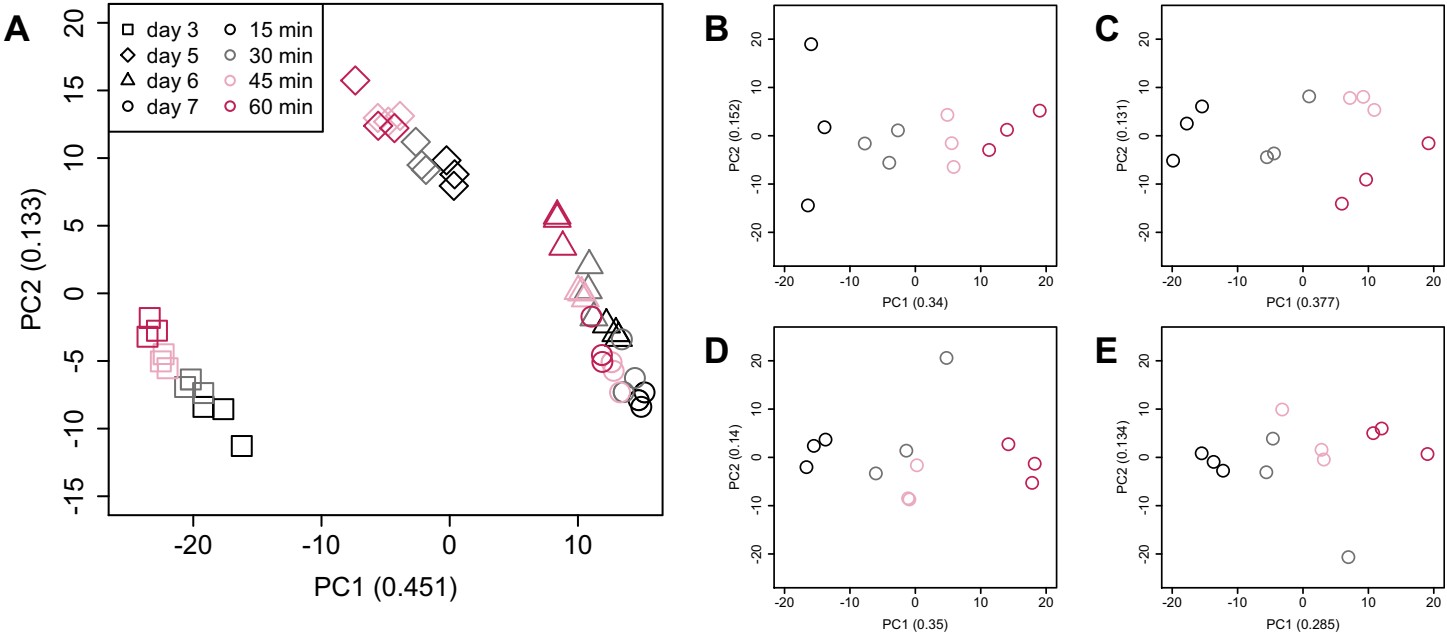

**Figure 7 Score plot of principal component analysis (PCA) using PC1 and PC2.** PCA plot of the 48 samples included in the study based on data from genes identified as differentially expressed at least once in DEG analyses. PC1 and PC2 appeared to be associated with the processing lag time of each cultivation day. (A) All 48 samples. (B), (C), (D), and (E) Samples from days 3, 5, 6, and 7, respectively. Legend is the same as Fig. 4.

the processing lag time of each cultivation day (Fig. 7A). PCA for each cultivation day also appeared to be associated with the processing lag time (Figs. 7B–7E). In the dendrogram (Fig. 8), we identified three major clusters associated with cultivation day: cluster 1 included all day 3 samples; cluster 2 included all day 5 samples; cluster 3 included all days 6 and 7 samples. It was consistent with the PCA (Fig. 7) that samples of days 6 and 7 were not clearly separated in the dendrogram. Then, each sub-cluster was associated with the processing lag time. For example, cluster 3, which included days 6 and 7 samples, has three sub-clusters: cluster 3–1 included 15 and 30 min samples; cluster 3-–2 included 30 and 45 min samples; cluster 3–3 included 45 and 60 min samples. The influences of the processing lag time on the expression of transcriptome increased with cultivation day.

# DISCUSSION

To evaluate the influence of differences in the processing lag time of cell samples (*i.e.*, time until the addition of RNA lysis buffer) on RNA-Seq, we examined gene expression changes when RNA lysis buffer was added at various times after the harvest of CHO cells. Regardless of the number of days of cell cultivation (3, 5, 6, and 7 days) or the processing lag time (15, 30, 45, and 60 min), we obtained high-quality total RNA and high-quality sequence reads from all samples. The results of the value of Shannon's information entropy (Fig. 3), PCA (Fig. 4), and HCA (Fig. 5) showed that, for each cultivation day, the global trends of gene expression were similar between the samples with various processing lag times. In contrast, the expression levels of specific genes were affected by the processing lag time of the samples, despite the total RNA being of high quality. The number of

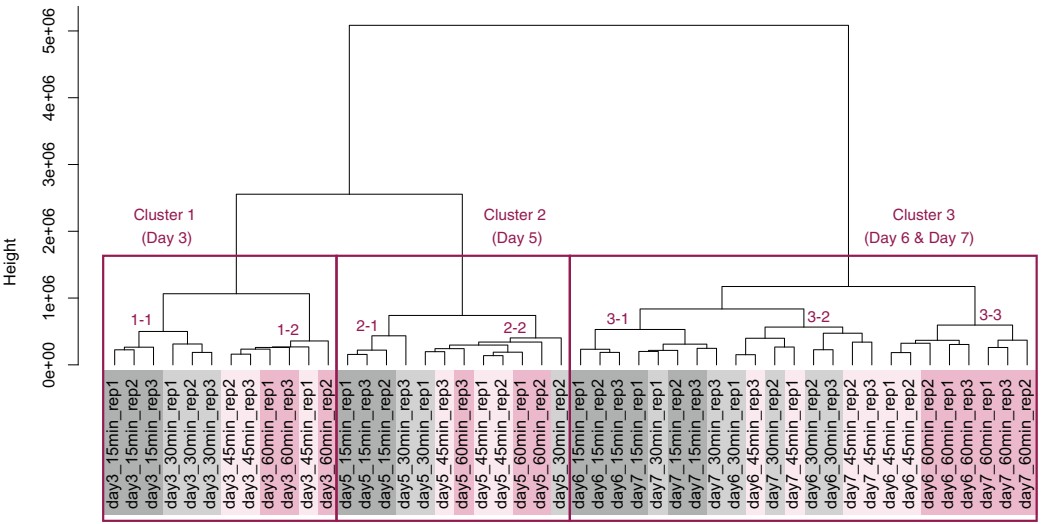

**Figure 8 Dendrogram resulting from hierarchical clustering analysis (HCA).** HCA was performed based on data from genes identified as differentially expressed at least once in DEG analyses. Each sample is indicated by the number of days of cultivation, the number of minutes of the processing lag time, and the technical replicate number. The sample names are as described in the Fig. 5 legend. There are three major clusters associated with cultivation day and seven sub-clusters associated with the processing lag time.                                                                  

common DEGs including apoptosis-associated genes increased as the processing lag time increased (Fig. 6, Table 1). In the dendrogram of the HCA performed for genes identified as differentially expressed at least once in DEG analyses (Fig. 8), each cluster consisted of several sub-clusters associated with the processing lag time. Especially, cluster 3 including the samples of days 6 and 7 separated into three sub-clusters associated with the processing lag time rather than cultivation day. Thus, we conclude that RNA lysis buffer should be added within 30 min after cell harvest. Otherwise, changes in the expression of some genes, especially genes associated with apoptosis, will occur.

Several studies have reported that the use of RNA samples with RIN values of around 5.0–7.0 does not negatively affect estimates of the gene expression profile (*e.g.*, *Fan et al., 2016*; *Shen et al., 2018*) or *de novo* assembly (*Kono et al., 2016*). However, our results indicated that even if using high-quality RNA (RIN > 9.1) for RNA-Seq, the gene expression levels of specific genes were significantly different between samples that share the same cultivation day but have different processing lag times. For instance, three genes of the AP-1 transcription factor complex (Fos, Jun, and Atf4) were detected as DEGs (Table 1), with higher expression as the processing lag time increased. AP-1 transcription factors regulate the expression of target genes involved in a wide range of cellular processes, such as proliferation, survival, and death. Fos is regarded as an early response gene to both biochemical and mechanical stress in several cell types (*Peake et al., 2000*; *Kim et al., 2017*). In the case of CHO cells, shear stress (*Ranjan et al., 1996*), virus infection (*Zachos, Clements & Conner, 1999*), and cobalt chloride stimulation (*Gong et al., 2001*; *Zou et al., 2001*) have been known to induce AP-1 activation. Here, although the time spent on centrifuging and pipetting was equal for all samples (Fig. 1), the time placed in

media or $1 \times$ PBS at room temperature was longer for samples with the longer processing lag time, suggesting that expression of AP-1 transcription factors increased in response to $1 \times$ PBS stimulation and/or the conditions (temperature and $CO_2$ concentration) outside the incubator.

Because the above results demonstrate that the processing lag time influences the expression level of specific genes, it is necessary to know not only the quality of the data but also how the experiment was performed when acquiring RNA-Seq data from SRA and investigating the expression level of a specific gene. However, detailed information on experimental operations, such as the processing lag time, is usually not described in manuscripts.

In recent years, there has been growing focus on the importance of reporting detailed methods, resulting in several protocol repositories have been begun, such as Nature Protocol Exchange (https://protocolexchange.researchsquare.com/), Protocols.io (*Teytelman et al., 2016*), MethodsX (http://www.sciencedirect.com/science/journal/22150161), Journal of Visualized Experiments (JoVE, https://www.jove.com/), and others. These repositories are an open-access, free platform for sharing, and discussing protocols and can be used during research and manuscript submission to facilitate reproducibility. However, protocol repositories usually do not show the working time of the experiment, even if it is a time-sensitive experiment like RNA isolation. Detailed protocols of this study including the processing time can be found at Protocols.io (see Materials & Methods).

Insufficient metadata will reduce the value of sequencing experiments by reducing the reproducibility of the study and its reuse for other analyses (*Stevens et al., 2020*). There are metadata standards for RNA-Seq such as the MINISEQE (Minimum Information about a high-throughput Nucleotide Sequencing Experiment), the ENCODE (Encyclopedia of DNA Elements) standards, or the International Human Epigenome Consortium (IHEC) metadata model. By following those metadata standards, or the checklist system for different types of data implemented by the European Nucleotide Archive (ENA), experimental details of RNA-Seq, such as the length of the RNA population being selected and whether rRNA populations were removed before library prep, can be described without missing. Furthermore, the BioSamples database at EMBL-EBI does not restrict the types of attributes used to describe sample metadata (*Courtot et al., 2018*), allowing submitters to choose attributes freely. However, those metadata standards for RNA-Seq do not have attributes corresponding to the working time of RNA isolation. The metadata associated with this study including the processing time of RNA samples described according to the ENCODE standards can be obtained from SRA Run Selector (see Material & Methods).

Although efforts have been made regarding the description to improve the reproducibility of experiments, it is difficult to describe the time required for individual steps and the number of pipetting operations in protocols or metadata. In the future, it would be ideal to employ humanoid robots that can use pipettes, vortex mixers, incubators, refrigerators, and centrifuges for experiments. A robotic crowd (or cloud) biology laboratory (RCBL; *Yachie, Robotic Biology Consortium & Natsume, 2017*) makes it possible

to implement materials and methods complementary and robustly (*Ochiai et al., 2020*). The development of a humanoid robotic platform for sample processing would contribute to accurate recording and improve reproducibility of experiment operations.

## CONCLUSIONS

In this study, we showed that differences in the time lag from harvesting cells to protecting RNA affect the expression level of specific genes in RNA-seq experiments. As discussed above, the data submitter needs to provide detailed information of the experiment together with the sequence reads. The database user needs to be able to investigate the validity of the data including potential problems with how the experiment was performed when acquiring HTS data from the database. Expression levels of the 23 DEGs identified in this study could potentially be used to help check whether the RNA of biological samples was properly treated.

## ACKNOWLEDGEMENTS

I am grateful to Dr. N. Ogata for his assistance with the experiments and useful discussions.

### Funding
The authors received no funding for this work.

### Competing Interests
Tomoko Matsuda is an employee of a commercial company: Nihon BioData Corporation.

### Author Contributions
- Tomoko Matsuda conceived and designed the experiments, performed the experiments, analyzed the data, prepared figures and/or tables, authored or reviewed drafts of the paper, and approved the final draft.

### Data Availability
The CHO-S cell sequences described here are available at Sequence Read Archive (SRA)/European Nucleotide Archive (ENA)/DDBJ Sequence Read Archive (DRA): DRA006016.

### Supplemental Information
Supplemental information for this article can be found online at http://dx.doi.org/10.7717/peerj.11875#supplemental-information.

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
