# Peer review of "Importance of experimental information (metadata) for archived sequence data: case of specific gene bias due to lag time between sample harvest and RNA protection in RNA sequencing"

_PeerJ, doi:10.7717/peerj.11875_

## Round 0.1 · original submission · Major Revisions

Your manuscript has been reviewed by two experts in the field. Although both found it interesting and that it made some important points, there were several major issues that should be addressed in a revised manuscript.

[]

·

Basic reporting

The english used, while clear, is very verbose so sentences can be challenging to understand. You may wish to consider reviewing the general language used to improve it's readability

e.g

lines 18-21

Huge amounts of data produced by high-throughput sequencing technologies have been submitted into the Sequence Read Archive (SRA) for reuse by repository users. However, it may not be possible to clarify the validity of the sequence data without metadata containing experiment-specific information.

would be clearly written as

Large volumes of high-throughput sequencing data have been submitted to the Sequencing Read Archive (SRA). The lack of experimental metadata associated with the data makes reuse and understanding data quality very difficult.

Line 47 there is no citation for the SRA such as https://doi.org/10.1093/nar/gkq1019

Line 59 For example, this author has observed that the time taken to process the same number of RNA samples was 15 min for one technician and 60 min for another technician in the same laboratory;

This is an anecdotal assertion without any support. It would be good to see this sentence strengthened with evidence or soften to a general trend

Experimental design

The experimental design is strong for the hypothesis being tested, that longer RNAseq processing times lead to changes in the observed gene expression profiles, especially for genes linked to apotosis

The authors should consider if they could share their protocol more widely, in a structure which is easier to reproduce via service such as protocols.io

https://www.protocols.io/

Validity of the findings

This paper identifies a clear challenge with the public sequences databases and proposes an experiment to identify the scale and nature of the challenge.

The way the results are presented in the paper make it difficult to see how large the difference is when lag time is considered and the fact the authors added cultivation time to this experiment to has somewhat clouded the results.

It would be good to see the authors think about focusing some of their consideration on a single cultivation time and assessing differences just in processing time in more detail to give greater evidence for their core assertion that processing time has a strong impact on the gene expression profiles derived. Figure 3 either adding an extra panel or a different visualisation might help here.

Additional comments

Line 150 there is a distinct lack of metadata associated with the data submission to DDBJ DRA006016 which given the assertion of the paper that such metadata is required to reanalyse data seems like a significant oversight

The authors don't demonstrate any awareness of existing metadata standards for RNAseq data such as the MINISEQE recommendations from the Functional Genomics Data Society, the ENCODE standards or the Livestock standards from the Functional Annotation of Animal Genomes consortium

http://fged.org/site_media/pdf/MINSEQE_1.0.pdf
https://genome.ucsc.edu/ENCODE/protocols/dataStandards/ENCODE_RNAseq_Standards_V1.0.pdf
https://www.ncbi.nlm.nih.gov/pubmed/30311252

The recommendations in the discussion section lack specificity starting on line 261 the authors propose better metadata with submissions without stating what variables should be provided and what type of data each of those variables should contain.

The authors correctly identify that without a specific standard it can be difficult to compare between studies and researchers only tend to submit variables which they think are important but don't talk about possible solutions through systems like the standards described above or the checklist system that the ENA has implemented with this to help with this problem https://www.ebi.ac.uk/ena/submit/checklists

The authors assert that protocols reported via video are easier to reproduce that text protocols without providing any evidence of that assertion or any commentary about if text protocols can be improved themselves if an author would struggle to produce and upload a video protocol.

I cannot recommend this paper for publication in its current state. The problem it identifies is real and the experiment chosen can be used to demonstrate why one particular RNAseq processing variable is important but the paper is lacking any discussion of previous attempts to try and address this challenge not do they even follow their own recommendations and provide such detail in their own DDBJ data submission.

thank you

Laura Clarke, EMBL-EBI

·

Basic reporting

Overall the quality of English and reporting is very good. That said, I think there
are a number of points that need to be clarified before this paper can be accepted for
publication. I want to stress that this is not a matter of extra work, simply clarification.

Specifically it's not clear in figures 3 and 4 what correlations (or other measure) are being used. Are the correlations over the
technical replicates? Are they over the day by day data (as opposed to the different times between gathering the RNA samples)?

With respect to figure 5 the focus is on showing genes that are DE between different RNA gathering times but surely it is more
interesting to ask what genes are DE between days of measurement and then to see if that pattern is borne out between different
measuring times?

Finally, the focus on genes related to apoptosis seems arbitrary - there needs to be some motivation why this is being examined in
detail.

Experimental design

The overall experimental design is sound though I note the above comments on reporting and what is being computed should be noted here.

Validity of the findings

Overall the findings are valid, though I note the points above in the reporting section.

Additional comments

I think some clarifications need to be made here but it's nothing that can't be fixed.

---

## Round 0.2 · accepted · Accept

Thank you for the meaningful revisions that were made in response to the reviewers' comments. The manuscript has been improved because of the changes that were made, and I am happy to recommend acceptance.